# Impact of Pre-Operative Right Ventricular Response to Hemodynamic Optimization on Outcomes in Patients with LVADs

**DOI:** 10.3390/jcm11206111

**Published:** 2022-10-17

**Authors:** Ernesto Ruiz Duque, Paulino Alvarez, Yingchi Yang, Muhammad Khalid, Rupesh Kshetri, Ilias P. Doulamis, Anthony Panos, Alexandros Briasoulis

**Affiliations:** 1Division of Cardiovascular Diseases, University of Iowa Hospitals and Clinics, Iowa City, IA 52242, USA; 2Department of Surgery, The Johns Hopkins Hospital, Baltimore, MD 21287, USA

**Keywords:** LVAD, RV failure, hemodynamics

## Abstract

Background: Right ventricular failure (RVF) continues to affect patients supported with durable left ventricular assist devices (LVAD) and results in increased morbidity and mortality. Information regarding the impact of right ventricular response to pre-operative optimization on outcomes is scarce. Methods: Single-center retrospective analysis of consecutive patients who underwent first continuous flow LVAD implantation between 2006 and 2020. Patients with bi-ventricular support before LVAD or without hemodynamic data were excluded. Invasive hemodynamics at baseline and after pre-operative medical and/or temporary circulatory support were recorded. Patients were grouped in the following categories: A: No Hemodynamic RV dysfunction (RVD) at baseline; B: RVD with achievement of RV hemodynamic optimization goals; C: RVD without achievement of RV optimization goals. The main outcomes were right ventricular failure defined as inotropes >14 days after implantation, or postoperative right ventricular mechanical support, and all-cause mortality. Results: Overall, 128 patients were included in the study. The mean age was 58 ± 12.5 years, 74.2% were males and, 68.7% had non-ischemic cardiomyopathy. Hemodynamic RVD was present in 70 (54.7%) of the patients at baseline. RV hemodynamic goals were achieved in 46 (79.31%) patients with RVD and in all the patients without RVD at baseline. Failure to achieve hemodynamic optimization goals was associated with a significantly higher risk of RVF after LVAD implantation (adjusted OR 4.37, 95% CI 1.14–16.76, *p* = 0.031) compared with no RVD at baseline and increased 1-year mortality compared with no RVD (adjusted HR 4.1, 95% CI 1.24–13.2, *p* = 0.02) and optimized RVD (adjusted HR 6.4, 95% CI 1.6–25.2, *p* = 0.008).Conclusion: Among patients with RVD, the inability to achieve hemodynamic optimization goals was associated with higher rates of RV failure and increased 1-year all-cause mortality post LVAD implantation.

## 1. Introduction

Over the last 20 years left ventricular assist devices (LVAD) have consolidated as evidence-based therapy for selected patients with advanced heart failure to improve quality of life and increase survival [1]. LVAD hemocompatibility-related clinical adverse events such as pump thrombosis and stroke have decreased since the introduction of HeartMate 3 a third generation magnetically levitated centrifugal continuous-flow LVAD [2].

Right ventricular failure limits the positive impact of univentricular support and is associated with increased morbidity and mortality [3]. The incidence of RVF in the perioperative period following LVAD implantation varies from 20 to 40%, depending on the definition used [4]. Pre-operative risk stratification of right ventricular failure is challenging, and numerous risk scores and tools are available with poor to modest discrimination capacity [5].

Right atrial pressure (RAP) > 12 mm Hg, RAP to pulmonary capillary wedge pressure (PCWP) ratio > 0.59, and pulmonary artery pulsatility index < 1.85 (PAPi) are among the hemodynamic parameters and indexes that have been associated with increased risk of right ventricular failure after LVAD [6,7]. Information about the impact of improvement on those load-dependent parameters in short and long-term outcomes is scarce [8,9,10].

In this context, we sought to evaluate the impact of achieving hemodynamic optimization goals among patients undergoing LVAD implantation at our institution. 

## 2. Methods 

Single-center retrospective analysis of consecutive patients who underwent first continuous flow left ventricular assist device (LVAD) implantation between 2006 and 2020. Patients with bi-ventricular support before LVAD or with no hemodynamic data were excluded (Figure 1). The local institutional review board approved the study.

Patients admitted to UIHC due to end-stage heart failure were included in the data collection. 201 patients were approved for LVAD implantation from January 2006 to January 2020. Patients 18 years or older were included. Patients with CentriMag for Biventricular support, total artificial heart, ECMO pre-LVAD, pulsatile LVAD, no hemodynamic optimization pre-LVAD, LVAD exchange, and no data available were excluded.

Pre-operative right heart catheterization was performed in all patients and the use of intravenous diuretics, inotropes and temporary mechanical support was performed at the discretion of the treating physician. Pre-operative right ventricular dysfunction (RVD) was defined using hemodynamics pre-LVAD by at least two of the following [4,6,11,12]. CVP > 16 mm Hg, CVP/PCWP ratio > 0.63, RVSWi < 300 mm Hg × mL/m^2^ or PAPi < 1.85. Hemodynamic optimization was defined by at least two of the following: CVP < 16 mm Hg, CVP/PCWP ratio < 0.63, RVSWi > 300 mm Hg × mL/m^2^ or PAPi > 1.85 before LVAD implantation. Patients were grouped in the following categories: A: No Hemodynamic RV dysfunction (RVD) at baseline; B: RVD with achievement of RV hemodynamic optimization goals; C: RVD without achievement of RV optimization goals. The main outcomes were right ventricular failure defined as inotropes >14 days after implantation, postoperative right ventricular mechanical support, and all-cause mortality.

### Statistical Analysis

Continuous variables are presented as means ± standard deviations and analyzed by Student’s *t* test. Categorical variables are presented as frequency and percentage and analyzed with Fisher’s exact test. Patient follow-up time was calculated as the time from LVAD implant until death or the last follow up. Patients were censored at the time of heart transplantation. Overall survival was evaluated using a Cox model for death; variables included in the multivariable model were chosen a priori and included patient characteristics presented in Table 1. All significance tests were two-tailed and conducted at the 5% significance level. Statistical analysis was performed using STATA, (College Station, TX, USA).

## 3. Results

A total of 128 patients were included in the study (study flow diagram presented in Figure 1). The mean age was 58 ± 12.5 years, 74.2% were males and, 68.7% had non-ischemic cardiomyopathy. Hemodynamic RVD was present in 70 (54.7%) of the patients at baseline. Patients with RV dysfunction required more frequently renal replacement therapy post-LVAD (10% vs. 1% *p* = 0.02), mechanical circulatory support pre-LVAD (67.24% vs. 34.29%, *p* < 0.01). The most common MCS pre-LVAD was IABP (66% vs. 36% *p* < 0.01). Patients with RV dysfunction more frequent had INTERMACS 2 (55.17% vs. 27.14%, *p* < 0.01), high creatinine (1.56 vs. 1.31 mg/dL, *p* < 0.01, high aspartate transaminase (100.94 vs. 30.98, *p* < 0.01), and low albumin (3.48 vs. 3.68, *p* < 0.01). RV systolic function assessed by TAPSE was decreased in patients with RVD in 100% vs. 64% without RVD (*p* = 0.1). 

RV hemodynamic goals were achieved in 46 (79.31%) patients with RVD and in all the patients without RVD at baseline. The time for optimization (mean) was 7.4 days (±6.95) for patients with No-RVD and 9.70 days (±7.90) with RVD (*p* = 0.04). The time for optimization for Non optimized RVD was 10.25 days (±9.04), Optimized RVD 9.56 days (±7.69), and No-RVD 7.4 days (±6.95) (*p* = 0.21). Patients who achieved hemodynamic optimization goals were younger, more frequent percutaneous coronary intervention, and better kidney function (Table 1). Changes in right atrial pressure (RAP), PAPI, RA/PCWP and RVSWi are shown in Figure 1. During the hemodynamic optimization there was not significant difference of PCWP, CI by Thermodilution, PVR, TPG and DPG among patients with Non RVD, RVD optimized and RVD Non-Optimized. Table 2.

### 3.1. Hemodynamic Parameters

The right atrial mean pressure (RA mean) was 6.98 mm Hg (non-RVD), 8.73 mm Hg (RVD-Optimized) and 19.83 mm Hg (RVD-non optimized), respectively (*p* < 0.01). RA mean decreased by 4.51 points (Delta RA mean) from admission in patients with non-RVD, 11.32 points in patients with RVD-Optimized, and 1.5 points in patients with RVD-non optimized (*p* < 0.01 Table 2).

The pulmonary arterial pulsatility index (PAPi) was 6.38 (non-RVD), 4.14 (RVD-Optimized), and 1.25 (RVD-non optimized), respectively (*p* < 0.01). PAPi improved by 3.53 points (Delta PAPi) from admission in patients with non-RVD, 2.86 points in patients with RVD-Optimized, and 0.08 points in patients with RVD-non optimized (*p* = 0.12 Table 2).

The RA/PCWP ratio was 0.34 (non-RVD), 0.44 (RVD optimized), 0.9 (RVD-non-Optimized) (*p* < 0.01), respectively. RA/PCWP ratio improved by 0.10 points (Delta RA/PCWP) from admission in patients with non-RVD, 0.20 points in patients with RVD-Optimized, and worsened in patients with RVD-non optimized by 0.16 points (*p* < 0.01 Table 2).

Right Ventricle Stroke Work Index (RVSWi) was 730 (non-RVD), 608 (RVD optimized), 416 (RVD-non-Optimized), respectively (*p* < 0.01). RVSWi worsened by 12.5 points (Delta RVSWi) from admission in patients with non-RVD, improved by132 points in patients with RVD-Optimized, and worsened by 57 points in patients with RVD-non optimized (*p* < 0.01 Table 2).

### 3.2. Outcomes

RV Failure risk was significantly higher in patients with previous RV dysfunction (51.72% vs. 30% *p* = 0.01 Table 1). Univariate logistic regression analysis showed that non optimized RVD was associated with significantly higher risk of RVF after LVAD compared with no RVD at baseline (unadjusted Odds Ratio [OR] 4.38, 95% CI 1.17–16.4, *p* = 0.028). There was no difference in risk of RVF among those with nonoptimized versus optimized RVD (OR 2.16, 95% CI 0.57–8.23, *p* = 0.26). INTERMACS 2 (unadjusted OR 3.29, 95% CI 1.56–6.92, *p* = 0.002) was associated with an increased risk of perioperative RV failure post LVAD whereas age, type of cardiomyopathy, gender, renal function, bilirubin and albumin were not associated with RVF risk. After adjustment for INTERMACS category, the risk of RVF remained significantly increased among patients with non optimized RVD before LVAD (OR 4.37, 95% CI 1.14–16.76, *p* = 0.031) compared with no RVD.

RV failure post-LVAD increased significantly the risk for all-cause mortality at 12 months post-implant (unadjusted hazard ratio [HR] 5.56 95% CI 1.8–17, *p* = 0.003). Patients without RV dysfunction had similar 12-month mortality compared with optimized RV Dysfunction (HR 1.33, 95% CI 0.4–4.4, *p* = 0.64). Patients with Non-Optimized RV dysfunction had significantly higher mortality at one-year post-implant compared with optimized RV dysfunction (41.66%, vs. 8.69% *p* < 0.01, HR 5.57, 95% CI 1.5–20.8, *p* = 0.11, Table 3). Patients that failed to achieve hemodynamic optimization goals exhibited increased 1-year mortality (HR 4.1, 95% CI 1.24–13.2, *p* = 0.02, Figure 2) adjusted for age, INTERMACS category, type of cardiomyopathy, creatinine, total bilirubin and albumin, compared with no RVD at baseline. Additionally, adjusted all-cause mortality at one-year was significantly higher among non optimized RVD compared with optimized RVD patients (HR 6.4, 95% CI 1.6–25.2, *p* = 0.008)

## 4. Discussion

The main findings of our study are as follows: Hemodynamic right ventricular dysfunction is prevalent and associated with an increased risk of postoperative RV failure and mortality. Failure to achieve hemodynamic targets of RV optimization is associated with an increased risk of RV failure and higher 1-year mortality after LVAD implantation.

RV failure defined as the requirement for an RVAD or continued use of inotropes >14 days after implantation is usually the most common cause of early postoperative HF which occurs in as many as 30% of patients in recent randomized controlled trials [2,5,6]. Preoperative RV dysfunction, excessive volume resuscitation and transfusion, perioperative RV injury due to prolonged cardiopulmonary bypass time, misalignment of the LVAD inflow cannula, and LVAD thrombosis are the main causes of early RV failure, which is associated with increased length of stay, major bleeding, renal failure, need for reoperation, poor outcomes post-transplant and increased mortality [2]. Late onset RV failure occurs in approximately 10% of LVAD recipients and portends a worse prognosis [5,6]. Late-onset HF after LVAD implantation can be due to primarily RV, primarily left-sided or mixed biventricular failure from LVAD-associated or non–LVAD-associated causes. In the setting of LVAD support, LV unloading decreases LV size, and leads to distortion of the geometry of the RV, resulting in septal bowing. This septal bowing can in turn cause obstruction to RV outflow as well as decreased RV stroke volume and worsening tricuspid regurgitation [5]. However, LVAD support also decreases PA pressures and RV afterload and results in augmented RV performance. Support with continuous flow LVADs has been found to improve RV function and decrease RA pressures independent of device speed [6].

In the absence of other etiologies, progressive RV dysfunction is treated initially with device optimization under echocardiographic guidance of invasive hemodynamic evaluation and diuresis. The reinstitution of inotropic support may be necessary in cases of advanced RV dysfunction, but it is associated with a higher burden of arrhythmias, line infections, and thrombosis without robust improvement of patients’ symptoms. Therefore, the identification of patients who are at higher risk for postoperative RV failure should be an integral part of the pre-LVAD evaluation. Despite the availability of several different risk scores [9,10,11,12], RV failure after LVAD implantation has proved difficult to predict, and currently, there is no consensus on how best to predict early or late right heart failure in this population. The optimization of invasive hemodynamics before LVAD implantation is an approach frequently used in advanced heart failure centers to prevent adverse outcomes including RVF after device implantation. However, data are limited on the predictive value of preimplant hemodynamics for the risk of RVF after LVAD implantation. Recently, in a single-center study of LVAD recipients from Mayo Clinic [10] identified BUN > 40, RAP > 12, and RVSWI < 500 as parameters associated with increased risk of the primary outcome of RV failure events, with gradually increasing risk as risk factors accumulated. In this study no specific strategy of optimization was implemented a priori but decisions were based on multidisciplinary discussions and achievement of widely accepted cut-offs for right atrial and pulmonary arterial pressure. In our study, patients were categorized according levels of RVD and response to optimization to examine the hypothesis that failure to optimize hemodynamics results in inferior post implantation outcomes. Although, the majority of patients either had acceptable RV hemodynamics or optimized before surgery, the minority of patients with persistently abnormal RV hemodynamics exhibited higher risk of RVF and mortality at one year. These findings highlight not only the importance of RVD in LVAD candidate selection but also the value of active optimization effort to improve short and long term outcomes. Our findings are hypothesis generating due to the small sample size but given the increasing number of institutions practicing optimization strategies before LVAD, there is accumulating evidence to support prospective clinical trial for incorporation of specific algorithms in clinical practice. Furthermore, our analysis supports the use of invasive hemodynamics as the principal tools for assessing RV function, guiding patient selection and peri-operative course after LVAD implantation.

The main limitation of our study is its retrospective design and sample size. An additional limitation of this study is that it represents the experience of one single center, where most patients were implanted with HeartMate II LVAD as DT, thus results might not be generalized to other health systems with different patient characteristics and using different continuous-flow LVADs.

## 5. Conclusions

Pre-LVAD implantation hemodynamic RV dysfunction is associated with RVF after LVAD implantation. Patients with RVD who achieved hemodynamic optimization goals had a lower risk of post LVAD RV failure and 1-year mortality. Initial response to hemodynamic optimization would be a potential risk stratification tool to assist in decision making before LVAD implantation.

## Figures and Tables

**Figure 1 jcm-11-06111-f001:**
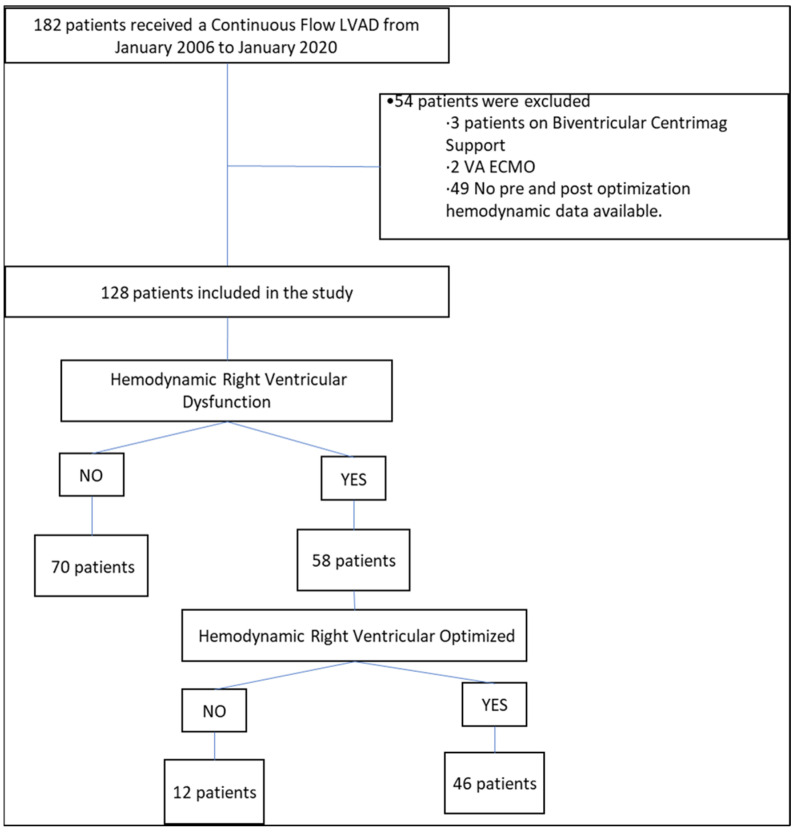
Flow diagram of included patients.

**Figure 2 jcm-11-06111-f002:**
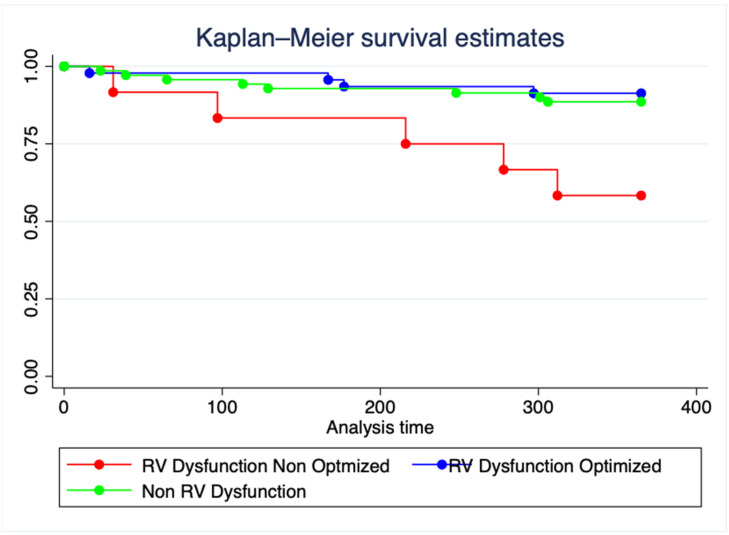
One-year all-cause mortality according to preimplantation RV hemodynamics. Cox regression model adjusted by age, type of cardiomyopathy, INTERMACS, creatinine, total bilirubin, and albumin.

**Table 1 jcm-11-06111-t001:** Pre-LVAD baseline characteristics among patients with RV dysfunction vs. without RV Dysfunction.

Pre-LVAD	Total	SD/%	No RV Dysfunction	SD/%	RV Dysfunction Optimized	SD/%	RV Dysfunction No Optimized	SD/%	*p*-Value
Demographic	128	100	70	54.7	46	35.9	12	9.4	
Age, y	57.96	12.5	60.31	11.3	54.5	14.57	57.58	7.02	0.04
Male	95	74.2	53	75.71	34	73.91	8	66.67	0.8
Ischemic Cardiomyopathy	40	31.3	26	37.14	12	26.09	2	16.67	0.23
Hypertension	53	41.4	33	47.14	15	32.61	5	41.67	0.29
Diabetes	60	46.9	34	48.57	22	47.83	4	33.33	0.61
COPD	19	14.8	12	17.14	6	13.04	1	8.33	0.66
Chronic Kidney Disease	45	35.2	27	38.57	15	32.61	3	25.00	0.59
Stroke	9	7.0	3	4.29	6	13.04	0	0.00	0.11
Coronary Artery Disease	57	44.5	37	52.86	16	34.78	4	33.33	0.11
CABG	18	14.1	11	15.71	5	10.87	2	16.67	0.73
PCI	40	31.3	28	40.00	11	23.91	1	8.33	0.03
Peripheral Vascular Disease	6	4.7	3	4.29	3	6.52	0	0.00	0.61
VF/VT	53	41.4	28	40.00	21	45.65	4	33.33	0.69
AFib/Aflutter	49	38.3	25	35.71	20	43.48	4	33.33	0.65
ICD	104	81.3	58	82.86	37	80.43	9	75	0.8
Laboratory									
Hemoglobin	11.32	1.93	11.4	1.87	11.2	2.08	10.87	1.78	0.63
WBC	8.46	3.17	8.2	2.87	8.8	3.44	8.5	3.91	0.59
Platelets	199.7	70.14	198	68.48	204	75.12	186	63.28	0.73
Na+	135.5	4.77	136	4.38	134	4.9	135	5.64	0.05
K+	4.02	0.51	3.9	0.48	4	0.57	4.2	0.31	0.28
Creatinine	1.42	0.53	1.3	0.46	1.5	0.59	1.6	0.55	0.02
BUN	29.81	17.21	27.5	13.78	30.3	17.11	41	29.28	0.03
AST	62.93	208	30	27.82	117	340.2	38	35.76	0.08
ALT	84.71	299	51	171	142	447.3	54	66.78	0.26
Total Bilirubin	1.03	0.66	0.95	0.66	1.15	0.68	1.6	0.6	0.27
Albumin	3.59	0.5	3.6	0.49	3.4	0.48	3.5	0.52	0.09
INR	1.61	1.09	1.5	0.92	1.7	1.41	1.4	0.45	0.47
Echocardiogram									
LVEF mean (SD)	13.2	10.43	12	11.66	14	8.93	17	6.89	0.23
LVEDd mean (SD)	6.41	1.7	6.21	1.82	6.6	1.42	6.6	1.35	0.3
Mitral Regurgitation (MR)									0.78
No MR	15	11.7	9	12.86	4	8.70	2	16.67	0.68
Mild MR	48	37.5	27	38.57	16	34.78	5	41.67	0.87
Moderate MR	48	37.5	25	35.71	18	39.13	5	41.67	0.89
Severe MR	17	13.3	9	12.86	8	17.39	0	0.00	0.28
Tricuspid Regurgitation (TR)									0.5
No TR	14	10.9	9	12.86	5	10.87	0	0.00	0.42
Mild TR	83	64.8	48	68.57	28	60.87	7	58.33	0.62
Moderate TR	29	22.7	12	17.14	12	26.09	5	41.67	0.13
Severe TR	2	1.6	1	1.43	1	2.17	0	0.00	0.85
LVAD Brand									0.76
HVAD	11	8.6	7	10.00	3	6.52	1	8.33	
HeartMate2	78	60.9	40	57.14	29	63.04	9	75.00	
HeartMate3	39	30.5	23	32.86	14	30.43	2	16.67	
LVAD Indication									0.61
Bridge to Transplant	73	57.0	41	58.57	24	52.17	8	66.67	
Destination Therapy	55	43.0	29	41.43	22	47.83	4	33.33	
INTERMACS									
INTERMACS 2	51	39.8	19	27.14	27	58.70	5	41.67	0.002
INTERMACS 3	64	50.0	40	57.14	18	39.13	6	50.00	0.16
INTERMACS 4	10	7.8	9	12.86	0	0.00	1	8.33	0.04
INTERMACS 5	3	2.3	2	2.86	1	2.17	0	0.00	0.83
Home Inotrope Pre-LVAD	59	46.1	29	41.43	23	50.0	7	58.33	0.44
IABP pre-LVAD	63	49.2	25	35.71	31	67.4	7	58.33	<0.01
IMPELLA pre-LVAD	4	3.1	2	2.86	2	4.3	0	0.00	0.72
Post-Surgery									
Chest open	20	15.6	8	11.43	10	21.74	2	16.67	0.32
ECMO post LVAD	1	0.8	1	1.43		0.00		0.00	0.65
Tracheostomy post LVAD	7	5.5	3	4.29	1	2.17	3	25.00	<0.01
CRRT post LVAD	7	5.5	1	1.43	4	8.70	2	16.67	0.04

LVAD: Left ventricular assist device; SD: Standard deviation; RV: Right ventricular; COPD: Chronic obstructive pulmonary disease; CABG: Coronary artery bypass grafting; PCI: Percutaneous coronary intervention; VF: Ventricular fibrillation; VT: Ventricular tachycardia; Afib: Atrial fibrillation; Aflutter: Atrial flutter; ICD: Implantable cardiac defibrillator; WBC: White blood cells; BUN: Blood urea nitrogen; AST: Aspartate transaminase; ALT: Alanine transaminase; INR: International Normalized Ratio; LVEF: Left ventricular ejection fraction; LVEDd; Left ventricular end diastolic diameter; HVAD: Heart ventricular assist device; IABP: Intra-aortic balloon pump; INTERMACS: Interagency Registry for Mechanically Assisted Circulatory Support; ECMO: Extra-corporeal membrane oxygenation; CRRT: Continuous renal replacement therapy.

**Table 2 jcm-11-06111-t002:** Pre-LVAD hemodynamic optimization among patients without RV dysfunction, RV Dysfunction and optimized hemodynamics and RV Dysfunction without optimized hemodynamics.

Optimized Hemodynamic	Total	SD	No RV Dysfunction	SD	RV Dysfunction Optimized	SD	RV Dysfunction Non-Optimized	SD	*p*-Value
RA mean	8.82	5.35	6.98	4.16	8.73	3.67	19.83	3.66	<0.01
Delta RA mean	6.67	6.2	4.51	5.1	11.32	5.09	1.5	5.05	<0.01
PCWP	20.87	6.71	21.02	7.16	19.89	6.17	23.75	5.42	0.2
PVR	2.6	1.68	2.67	1.98	2.36	1.11	3.11	1.56	0.34
CI Thermo	2.37	0.55	2.35	0.47	2.44	0.66	2.36	0.51	0.5
PAPI	5.09	5.58	6.38	8.87	4,14	5.26	1.25	0.35	<0.01
Delta PAPI	2.97	5.46	3.53	6.01	2.86	5.09	0.08	0.22	0.12
RA/PCWP	0.43	0.25	0.34	0.19	0.44	0.15	0.9	0.37	<0.01
Delta RA/PCWP	0.11	0.24	0.1	0.19	0.2	0.18	−0.16	0.4	<0.01
RVSWI	657	298	730	315	608	258	416	152	<0.01
Delta RVSWI	35.36	277	12.5	298	132	243	57	157	0.01
TPG	11.81	6.61	11.9	7.71	11.2	5.12	13	4.47	0.67
DPG	1.66	5.06	1.24	5.66	1.93	4.23	3	4.23	0.46

PAC: Pulmonary Artery Catheter. RA: right atrial mean pressure. PASP: Pulmonary arterial Systolic Pressures. PADP: Pulmonary Arterial Diastolic Pressures. PAMP: Pulmonary Arterial Mean Pressures. PCWP: Pulmonary Capillary Wedge Pressure. PVR: Pulmonary Vascular Resistant Woods Units. CO Fick: Cardiac Output by Fick. CO Thermo: Cardiac Output by Thermodilution. CI Fick: Cardiac Output Indexed by Fick. CI Thermo: Cardiac Output Indexed by Thermodilution. PAPI: Pulmonary Artery Pulsatility Index. RA/PCWP: right atrial mean pressure/ Pulmonary Capillary Wedge Pressure ratio. RVSWI: Right Ventricle Work Wall Index. TPG: Transpulmonary Gradient. DPG: Diastolic Transpulmonary Gradient.

**Table 3 jcm-11-06111-t003:** Right Ventricle outcomes after LVAD implantation.

	Total	%	Non-RV Dysfunction	%	RV Dysfunction Optimized	%	RV Dysfunction Non-Optimized	%	*p*-Value
Right Ventricular Failure	51	39.8	21	30.00	22	47.8	8	66.67	0.02
RVAD	16	12.5	6	8.57	8	17.4	2	16.67	0.34
Inotrope > 14 days	48	37.5	20	28.57	21	45.7	7	58.33	0.05
Inotrope > 21 days	31	24.2	15	21.43	12	26.1	4	33.33	0.63
12 Months Mortality	17	13.3	8	11.43	4	8.7	5	41.67	<0.01
24 Months Mortality	28	21.9	17	24.29	6	13.0	5	41.67	0.07

RV: Right ventricular; RVAD: Right ventricle assist device.

## Data Availability

Data available upon request.

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
