# Peer review of "Impact of Pre-Operative Right Ventricular Response to Hemodynamic Optimization on Outcomes in Patients with LVADs"

_jcm, 2022, doi:10.3390/jcm11206111_

Round 1
Reviewer 1 Report
In the present study, Drs Duque, Alvarez and colleagues, evaluated the impact of pre-operative right ventricular response to hemodynamic optimization on outcomes in patients with LVADs
The manuscript is clear, well written and interesting.
A few major issues need to be addressed:
1. In the Introduction section pre-last paragraph (page 2 line 4): would be helpful and more informative to add the values of RA pressures, RAP to pulmonary capillary wedge pressure (PCWP) ratio, and pulmonary artery pulsatility index (PAPi) that have been associated with increased risk of post LVAD RV failure and not only cite the ref.
2. In the Methods section:
a. References should be attached to the relevant RVD values.
b. What was the period of time between the hemodynamic assessment and surgery?
c. Were there only two hemodynamic studies performed per each patient? If no, what determined the decision to perform the in-between hemodynamic studies?
d. What inotropic protocol was used?
e. If available, GGT & AP values would be interesting as these values reflect liver congestion as a marker of significant RVD.
f. In table 1: By what is written, decrease RV systolic function by TAPSE was observed in 77% of the group of patients with "No RV dysfunction" this contradiction should be cleared either by clarifying that the definition of RVD was based only on hemodynamic parameters or by changing the values in the table.
g. Elaborate shortly on the indications for "Home inotropes" in patients with no RVD.. Why was the LVAD implantation delayed? What was the dose and nature of the "home inotrope"?
3. In the Results section:
a. page 5 line 5: "…mechanical circulatory support pre-LVAD (67.24 %
vs 34.29 %, p<0.01), and IABP was the most common MCS (66% vs 36% p<0.01)." This sentence is unclear.
b. Wherever appears, no RVD means no hemodynamic RVD? What about those 30% with no hemodynamic RVD: how was the RV function by echo?
c. How were the ethical issues of implanting an LVAD in patients with non-optimized RV function solved?? Were there any patients that LVAD implantation was declined if RV optimization was not achieved??
4. In the discussion section:
a. In the first sentence: "The salient findings of our study are as follows: Hemodynamic right ventricular dysfunction is prevalent and associated with an increased risk of postoperative RV failure and mortality: No novelty here.
b. In the second part of that first paragraph: "Failure to achieve hemodynamic targets of RV optimization is associated with an increased risk of RV failure and higherr 1-year survival after LVAD implantation…" The spelling mistake at higher should be corrected. And probably a human error: probable should read mortality and not survival.
Author Response
Dear Editor and Reviewers,
Thank you for your kind note dated September 21st, 2022 on our paper “JCM-1922105, “Impact of pre-operative right ventricular response to hemodynamic optimization on outcomes in patients with LVADs”. We are very grateful for your comments and the thoughtful suggestions from the reviewers. Based on them, we have made careful modifications to the original manuscript and carefully proof-read it. We believe that our paper has greatly improved, and we hope it has reached the required standards for Journal of Clinical Medicine.
Once again, we acknowledge your comments, which are valuable in improving the quality of our manuscript.
Yours sincerely,
Alexandros Briasoulis MD
University of Iowa. 200 Hawkins Drive, Iowa City, IA 52245.
Telephone: (319) 678-8418. Fax: (319) 353-6443.
E-mail: alexbriasoulis@gmail.com
We thank the Editors for their time and efforts to improve our manuscript.
Reviewer: 1
Comment: The manuscript is clear, well written and interesting.
A few major issues need to be addressed: In the Introduction section pre-last paragraph (page 2 line 4): would be helpful and more informative to add the values of RA pressures, RAP to pulmonary capillary wedge pressure (PCWP) ratio, and pulmonary artery pulsatility index (PAPi) that have been associated with increased risk of post LVAD RV failure and not only cite the ref.
Response: RAP, PCWP and PAPi cut off number as a predictor for RV failure post LVAD were reviewed according with the reference used and mentioned as suggested.
Comment: In the Methods section: References should be attached to the relevant RVD values Response: References were added.
Comment: What was the period of time between the hemodynamic assessment and surgery?
Response: The time for optimization (mean) was 7.4 days (+/- 6.95) for patients with No-RVD and 9.70 days (+/- 7.90) with RVD (p=0.04). The time for optimization for Non optimized RVD was 10.25 days (+/- 9.04), Optimized RVD 9.56 days ( +/- 7.69), and No-RVD 7.4 days (+/- 6.95) (p=0.21).
Comment: Were there only two hemodynamic studies performed per each patient? If no, what determined the decision to perform the in-between hemodynamic studies?
Response: Thank you for the thoughtful insight. It was a continue invasive hemodynamic assessment indeed. The PA Cath was inserted, and hemodynamics were used daily to make decision about optimization. We decided to use the baseline hemodynamics and the last hemodynamics before surgery.
Comment: What inotropic protocol was used?
Response: The Inotrope used was up to the Physician rounding in the CVICU during the admission. Usually, borderline blood pressure and abnormal kidney function would lead the physician to use dobutamine. In the case of pulmonary hypertension, the use of milrinone was opted.
Comment: If available, GGT & AP values would be interesting as these values reflect liver congestion as a marker of significant RVD.
Response: We agree with the reviewer; however, the respective data is not available.
Comment: In table 1: By what is written, decrease RV systolic function by TAPSE was observed in 77% of the group of patients with "No RV dysfunction" this contradiction should be cleared either by clarifying that the definition of RVD was based only on hemodynamic parameters or by changing the values in the table.
Response: We would like to thank the reviewer for pointing out this issue. RVD definition was only a hemodynamic assessment. The echocardiogram information was collected any time during 3 months before or during the admission. TAPSE has been a weak predictor of RVF post LVAD. The RV function might have improved after optimization, but we did not evaluate TAPSE post optimization. We agree, No-RVD along decrease RV systolic function pre LVAD it is contradictory. The risk of RVF post LVAD in our cohort with TAPSE <17 mm was (OR 1.91 95% CI 0.76-4.99, P = 0.18). RVD pre LVAD defined by hemodynamics showed higher risk for RVF post LVAD (OR 2.5 95% CI 1.21- 5.16, p=0.01). Thus, TAPSE was removed from Table 1 per reviewer’s suggestion.
Comment: Elaborate shortly on the indications for "Home inotropes" in patients with no RVD. Why was the LVAD implantation delayed? What was the dose and nature of the "home inotrope"?
Response: Patients with HOME inotrope had low CI < 1.8, PCWP > 15 mmHg regardless of RV hemodynamics parameters. Dobutamine dose in patients with (RVD 4.13 mcg/kg/min vs Non-RVD 3.59 mcg/kg/min), and Milrinone (0.21 mcg/kg/min vs 0.24 mcg/kg/min).
The LVAD implant date delayed was not controlled by the medical team. One reason was further optimization after PA cath numbers and/or OR time availability; however, which variable leaded the delay cannot be determined.
Comment: In the Results section: page 5 line 5: "…mechanical circulatory support pre-LVAD (67.24 %
vs 34.29 %, p<0.01), and IABP was the most common MCS (66% vs 36% p<0.01)." This sentence is unclear.
Response: Thank you for the comment. The IABP was the most common MCS used (66% vs 36%) The total MCS used includes Impella which represent a small percentage (Total 66% vs 36%).
Comment: Wherever appears, no RVD means no hemodynamic RVD? What about those 30% with no hemodynamic RVD: how was the RV function by echo?
Response: Indeed, No-RVD means no RV dysfunction assessed by hemodynamics. We did not take in consideration RV function (TAPSE) to classify RV dysfunction as we discussed above (Reviewer question 2-f). The echocardiogram was collected in a different time of hemodynamics. This might explain the discrepancy of the results.
Comment: How were the ethical issues of implanting an LVAD in patients with non-optimized RV function solved?? Were there any patients that LVAD implantation was declined if RV optimization was not achieved??
Response: This is a great point raised by the reviewer. This is a retrospective study where we analyzed the hemodynamics and dividing the groups accordingly. The decision to implant the LVAD was made before optimization. Nay patient was declined on Non optimized RVD group.
Comment: In the discussion section: In the first sentence: "The salient findings of our study are as follows: Hemodynamic right ventricular dysfunction is prevalent and associated with an increased risk of postoperative RV failure and mortality. No novelty here.
Response: We rephrased this sentence and removed the word “salient”.
Comment: In the second part of that first paragraph: "Failure to achieve hemodynamic targets of RV optimization is associated with an increased risk of RV failure and higherr 1-year survival after LVAD implantation…" The spelling mistake at higher should be corrected. And probably a human error: probable should read mortality and not survival.
Response: Thank you for highlighting this error. It has now been rectified.
Reviewer 2 Report
The present manuscript emphasizes the impact of pre-operative RV optimization on results after LVAD implantation. Their conclusion is that optimized RV function results in better survival after VAD implantation.
Although the number of patients is quite small and the results are mainly based on HM2 patients the message to improve RV function is clear.
Author Response
Dear Editor and Reviewers,
Comment: The present manuscript emphasizes the impact of pre-operative RV optimization on results after LVAD implantation. Their conclusion is that optimized RV function results in better survival after VAD implantation.
Although the number of patients is quite small and the results are mainly based on HM2 patients the message to improve RV function is clear.
Response: We would like to thank the reviewer for taking the time to review this paper and for these kind words.